

# Characterization of the airborne aerosol inlet and transport system used during the A-LIFE aircraft field experiment

Manuel Schöberl[1,2], Maximilian Dollner[1], Josef Gasteiger[1,a], Petra Seibert[3,4], Anne Tipka[1,3,b], and Bernadett Weinzierl[1]

[1]University of Vienna, Faculty of Physics, Aerosol Physics and Environmental Physics, 1090 Vienna, Austria
[2]University of Vienna, Vienna Doctoral School in Physics, 1090 Vienna, Austria
[3]University of Vienna, Department of Meteorology and Geophysics, 1090 Vienna, Austria
[4]University of Natural Resources and Life Sciences, Institute of Meteorology and Climatology, 1180 Vienna, Austria
[a]now at: Hamtec Consulting GmbH @ EUMETSAT, Darmstadt, Germany
[b]now at: International Data Centre, Comprehensive Nuclear-Test-Ban Treaty Organization, PO Box 1200, 1400 Vienna, Austria

*Correspondence to*: Bernadett Weinzierl (bernadett.weinzierl@univie.ac.at)

**Abstract.**

Atmospheric aerosol particles have a profound impact on Earth's climate by scattering and absorbing solar and terrestrial radiation and by impacting the properties of clouds. Research aircraft such as the Deutsches Zentrum für Luft- und Raumfahrt e.V. (DLR) Falcon are widely used to study aerosol particles in the troposphere and lower stratosphere. However, transporting a representative sample to the instrumentation inside the aircraft remains a challenge due to high airspeeds and changing ambient conditions. In particular, for high-quality coarse mode aerosol measurements, knowledge about losses or
enhancements in the aerosol sampling system is crucial. In this study, we present a detailed characterization of the Falcon aerosol sampling system. Aerosol number size distributions were measured during the A-LIFE field campaign simultaneously with in-cabin and out-cabin/wing-mounted instrumentation. Sampling efficiencies were derived for different true airspeed ranges by comparing the in-cabin and the out-cabin particle number size distributions during flight sequences with a major contribution of mineral dust particles in the coarse mode size range. Additionally, experimentally derived Stokes numbers
were used to calculate the cut-off diameter of the A-LIFE aerosol sampling system for different particle densities as a function of true airspeed. The results show that the velocity of the research aircraft has a major impact on the sampling of coarse mode aerosol particles with in-cabin instruments. For true airspeeds up to about 190 m s⁻¹, aerosol particles larger than about 1 µm are depleted in the sampling system of the Falcon during the A-LIFE project. In contrast, for true airspeeds higher than 190 m s⁻¹, an enhancement of particles up to a diameter of 4 µm is observed. For even larger particles, the enhancement effect at the
inlet is still present, but inertial and gravitational particle losses in the transport system get more and more pronounced which leads to a decreasing overall sampling efficiency. In summary, aerosol particles can either be depleted or enhanced at an aerosol inlet, whereas transport in sampling lines always leads to a loss of particles. Therefore, it is important to consider both, inlet and transport efficiency, when quantifying the sampling efficiency of an aerosol sampling system.



## 1 Introduction

Atmospheric aerosol particles play an important role in the changing climate system. Due to their ability to scatter and absorb solar and terrestrial radiation as well as their impact on cloud properties they can influence Earth's climate (Boucher et al., 2013). In recent years, there has been an increasing interest in studying aerosol particles directly in the atmosphere with an aircraft, equipped with aerosol instruments installed in the aircraft cabin, under the aircraft wings or at the fuselage (e.g. Wendisch and Brenguier, 2013; Weinzierl et al., 2017; Weinzierl et al., 2023, in prep.). Crucial for the in-cabin instrumentation

is the sampling system (consisting of inlets and sampling lines) aboard the aircraft which transports the ambient aerosol particles to the instruments. One of the greatest challenges of airborne measurements is to design and operate the sampling system such that biases in the measurements of aerosol properties are avoided despite facing different ambient conditions (e.g. varying true airspeed of the aircraft, changing temperature and pressure) (Baumgardner and Huebert, 1993; Wendisch et al., 2004). In particular, coarse mode aerosol particles (particle diameter > 1 µm) are affected by sampling effects due to their high

inertia which can result in an artificial depletion or enhancement (Hinds, 1999; Brockmann, 2011). Hence, a characterization of the sampling system[1] aboard any research aircraft is imperative.

An efficiency $\eta$ defines the change in particle number concentration before and after a certain part of the sampling system as a function of particle diameter $D_p$. In case of the sampling system aboard a research aircraft, changes in particle number

concentration can occur at the sampling system inlet and in the transport system. While the inlet efficiency describes particle losses ($\eta_{in} < 1$) or enhancements ($\eta_{in} > 1$) at the sampling system inlet, the transport efficiency $\eta_{tr}$ defines the particle depletion due to different loss mechanisms (e.g. gravitational settling, losses in bends) in the sampling lines, which connect the instrument with the inlet. In addition to the particle size, these efficiencies depend on various parameters including particle density, sampling system geometry (e.g. the inlet design, the number of bends in the transport system, the bend angle,

horizontal or vertical sampling lines), and flow velocity (Hinds, 1999; Brockmann, 2011). Hence, a sampling system can be characterized by the overall sampling efficiency $\eta_{sys}$ which is given by multiplying the inlet efficiency $\eta_{in}$ and the transport efficiency $\eta_{tr}$ (Brockmann, 2011):

$$\eta_{sys}\left(D_p\right) = \eta_{in}\left(D_p\right) \cdot \eta_{tr}\left(D_p\right) \tag{1}$$


A key aspect of the inlet efficiency is the ratio between the ambient air velocity $U_0$ and the stream velocity inside the inlet $U$ (Brockmann, 2011). A representative sample of ambient aerosol particles enters the inlet, if the ratio $U_0/U$ equals unity and the inlet is aligned in parallel to the flow direction. Sampling with $U_0/U=1$ is known as isokinetic sampling. For conditions with $U < U_0$, the sampling is called sub-isokinetic sampling: the ambient air streamlines diverge at the inlet entry and particles

---

[1] Note: in this study, we use the term „sampling system" to refer to both, inlet and sampling lines. We use the term „transport system" if we refer to the sampling lines between the inlet and the instrumentation.



with high inertia cannot follow the streamlines and are artificially enriched in the sampling system inlet. Super-isokinetic
       sampling is given for the opposite case ($U > U_0$) where the ambient air streamlines converge into the inlet. Thus, particles with
       high inertia are underrepresented in the sampling probe.

       Many research aircraft such as for example the DLR Falcon or the NASA DC-8 move with $U_0 > 100$ m s$^{-1}$ which is typically
greater than the stream velocity at the inlet entry (typically defined by the total volume flow needed by the in-cabin
       instrumentation). Nearly isokinetic sampling conditions can be accomplished by a diffuser which decelerates the ambient air
       before entering the sampling system (Seebaugh, 1991). Numerous studies investigated the sampling efficiency of such diffuser-
       type aerosol inlets with different approaches: For example, several airborne aerosol inlets (e.g., aboard the NASA DC-8
       research aircraft) were characterized experimentally by in-flight testing or comparisons of instrument data (Huebert et al.,
1990; Porter et al., 1992; Sheridan and Norton, 1998; McNaughton et al., 2007). Hermann et al. (2000) conducted wind-tunnel
       tests to characterize the aerosol inlet installed on a Boeing civil aircraft. A similar approach is described in Hegg et al. (2005)
       to determine the inlet's transmission efficiency of the CIRPAS Twin Otter research aircraft. Wilson et al. (2004) showed that
       coarse mode aerosol particles are enhanced in the sample flow and that the enhancement factor can be described as a function
       of the Stokes number. A more theoretical approach for various inlets is demonstrated in Krämer and Afchine (2003) by
comparing empirically derived equations from Belyaev and Levin (1974) to computational fluid dynamics (CFD) model
       results. In general, inlet systems are constantly being improved and renewed for different aircraft campaigns, for example in
       order to measure aerosol particles only in a certain size range (Dhaniyala et al., 2003; Perring et al., 2013).

       Here, we carry out a comprehensive characterization of the sampling system of the Falcon, which was used as the research
aircraft during the A-LIFE field campaign (Weinzierl et al., 2023, in prep.) and has been used for many aircraft missions in
       the past decades (e.g. SALTRACE (Weinzierl et al., 2017) or ACCESS (Moore et al., 2017)). Fiebig (2001) investigated the
       Falcon aerosol inlet's cut-off diameter $D_{p,50}$ (particle diameter at which the overall sampling efficiency equals 50 %) as a
       function of flight altitude during the Lindenberger Aerosol Characterisation Experiment (LACE 98; Ansmann et al., 2002),
       but the analysis was restricted to six flight sequences within the planetary boundary layer. In this study, we use the entire data
set of the A-LIFE mission for the characterization which covers the entire altitude range of the Falcon from the ground to about
       12 km.

       This paper is structured as follows: section 2 gives an overview of the A-LIFE aircraft field campaign, introduces the Falcon
       aerosol inlet and the A-LIFE aerosol transport system, and summarizes the derived aerosol number size distributions.
Furthermore, the methodology of the sampling system characterization is explained. The results are described in section 3 and
       contain the following key outcomes: classification of each A-LIFE flight sequence according to the sampling condition, cut-
       off diameters $D_{p,50}$ of the Falcon sampling system for a number of particle densities between 1.0 g cm$^{-3}$ and 2.6 g cm$^{-3}$, and



sampling efficiencies derived for different ranges of ambient air velocities $U_0$. The last two sections of this paper include the discussion of the results and the conclusion of this study.


## 2 Methodology

### 2.1 A-LIFE Field Campaign

The European Research Council (ERC) project *Absorbing aerosol layers in a changing climate: aging, lifetime and dynamics* (A-LIFE) conducted ground-based and airborne measurements to characterize mixtures of absorbing aerosol particles (in
particular mineral dust and black carbon) in the atmosphere. For A-LIFE, the DLR Falcon 20-E5 D-CMET research aircraft (hereafter simply Falcon) was used for the airborne measurements in spring 2017. During this time, the aircraft was based in Paphos, Cyprus, and provided access to dust layers originating from the Saharan or the Arabian Peninsula as well as to polluted aerosol layers (Weinzierl et al., 2023, in prep.).

In total, 22 research flights were conducted (including 4 transfer flights and 2 test flights) with a total amount of about 80 flight hours. For A-LIFE, a substantial number of aerosol instruments, a wind lidar and impactors for a subsequent analysis of filter samples were installed on the Falcon research aircraft. The resulting unique data set includes measurements of particle number size distribution in the size range of 10 nm up to 930 µm, aerosol optical properties, cloud condensation nuclei (CCN), and refractory black carbon (rBC) mass and mixing state. Additionally, a Rosemount five-hole pressure probe model 858 was
installed on the tip of the aircraft for meteorological measurements (hereafter referred to as the CMET system) (Bögel and Baumann, 1991). In order to define averaging time periods, the research flights were divided into flight sequences in which the Falcon flew at a constant altitude in quasi-homogeneous aerosol concentrations. Periods in which the Falcon flew in clouds were not considered, since cloud particles can produce artefacts at the inlet and thus artificially increase the number concentration (Murphy et al., 2004). The algorithm used for the detection of clouds is described by Dollner (2022) and Dollner
et al. (2023a, in prep.). Altogether, the A-LIFE data set was divided into 262 flight sequences.

To enable a consistent statistical analysis of the A-LIFE data set, an aerosol classification of all flight sequences was established. It was derived by considering in-situ measurements of coarse mode aerosol particles and refractory black carbon mass in combination with the Lagrangian particle dispersion model FLEXPART (Stohl et al., 1998, Seibert and Frank, 2004).
The model is driven by meteorological data from the European Centre for Medium-Range Weather Forecasts (ECMWF) and, in combination with emission data from the Copernicus Atmospheric Monitoring Service, provides quantitative information about the types of the measured aerosol and their origins. The aerosol classification scheme for A-LIFE (Weinzierl et al., 2023 in prep.) divides the data set into flight sequences with (174) and without (88) a primary contribution of mineral dust in the coarse mode size range (particle diameter > 1 µm). Further distinctions are made based on the degree of pollution (polluted,



moderately-polluted, clean). For this study, mainly sequences with a mineral dust contribution (hereafter abbreviated to dust sequences) are considered. Details about the A-LIFE aerosol classification can be found in Weinzierl et al. (2023, in prep.).

## 2.2 Falcon Aerosol Inlet and A-LIFE Aerosol Transport System

An aerosol sampling system consists of an inlet and a sampling line system connecting the inlet with the individual instruments. As outlined in equation (1), the overall sampling efficiency $\eta_{sys}$ is given as the product of the inlet efficiency $\eta_{in}$ and the

transport efficiency $\eta_{tr}$.

The Falcon aerosol inlet system, which was designed by Franz Schröder (formerly DLR), is depicted in Figure 1 and consists of four different inlets: isokinetic, forward-facing, sideward (not visible in Figure 1), and backward inlet (Fiebig, 2001). The flow plan for the aerosol transport system, which was installed and used during the A-LIFE campaign, can be seen as a

schematic in Figure S1 in the Supplemental Material. The majority of in-cabin instruments were connected to the isokinetic aerosol inlet. Therefore, we restrict the characterization to the isokinetic Falcon aerosol inlet.

### 2.2.1 Inlet System

Before the sample air reaches the isokinetic inlet, it passes through a diffusor in which the ambient air velocity $U_0$ is decelerated. The diffuser, which has the shape of a cone with an opening angle of $2\theta = 6.3°$, is mounted in parallel to the flight

direction with a distance of 30 cm to the aircraft's fuselage to be outside of the boundary layer of the aircraft. The deceleration of the ambient air is determined by the ratio of the cross-sectional areas at the end and at the beginning of the diffuser, which equals 7.1 in the case of the Falcon inlet. Furthermore, the sample air is transported through a bend of 90° with a radius of 19 cm, leading to the isokinetic inlet mounted perpendicular to the aircraft's fuselage (see Figure 1). The isokinetic inlet has an inner diameter of 4.572 mm and connects the inlet system to the sampling line system transporting the aerosol particles to the

individual instruments (see Figure S1 in the Supplemental Material). In total, the A-LIFE in-cabin instrumentation needed a volumetric flow between 17.87 and 22.83 l min$^{-1}$ at the isokinetic inlet (18.14-23.18 m s$^{-1}$ by considering the inlet's inner diameter). Note, the inlet efficiency $\eta_{in}$ presented in this study, refers to the efficiency of the parts consisting of the diffusor, the tube with the 90° bend, and the isokinetic inlet tube (i.e. the components shown in Figure 1).

As a measure of the ambient air velocity $U_0$, the true airspeed (hereafter abbreviated to $v_{TAS}$) is used. The $v_{TAS}$ is the velocity of the aircraft with respect to the surrounding air mass. The measurement of $v_{TAS}$ was provided by the CMET system. In Figure 2, a 2D histogram of the recorded $v_{TAS}$ during the A-LIFE campaign is depicted as a function of altitude, colour coded with the number of seconds spent in these conditions. The $v_{TAS}$ ranges roughly from 70 m s$^{-1}$ at ground level to 220 m s$^{-1}$ for altitudes up to nearly 12 km. With a flow velocity of 18.14-23.18 m s$^{-1}$ inside the isokinetic inlet and a deceleration of the ambient air

velocity by a factor of 7.1, isokinetic sampling conditions at the isokinetic inlet are established with a $v_{TAS}$ of approximately 129-165 m s$^{-1}$.









**Figure 1: Schematic of the Falcon aerosol inlet adapted from Fiebig (2001). The inlet was constructed by Franz Schröder (formerly DLR). The majority of the A-LIFE instrumentation was connected to the isokinetic inlet. In the diffuser, the ambient air stream velocity is reduced by a factor of 7.1. Figure reprinted from Fiebig (2001) with permission of Markus Fiebig, NILU.**








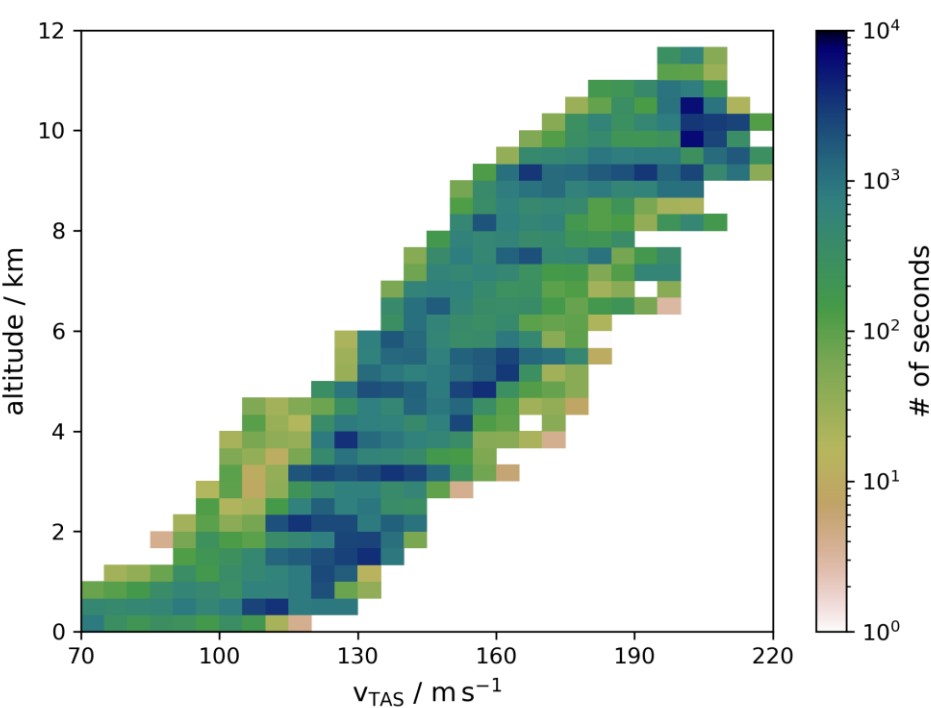

**Figure 2: 2D histogram of true airspeed values v$_{TAS}$ recorded by the CMET nose boom of the Falcon research aircraft as a function of altitude for every flight second during the A-LIFE mission.**



### 2.2.2 A-LIFE Aerosol Transport System

Sampling lines with a diameter of ¼ inch (6.35 mm) connected the isokinetic inlet with the in-cabin instrumentation and constituted the transport system aboard the Falcon. An overview of the instrumentation and the sampling line lengths is given in the Supplemental Material (Figure S1 and Table S1). Estimations of the transport efficiency $\eta_{tr}$ of coarse mode aerosol particles were computed considering sedimentation and bend losses with empirical equations from literature (see Thomas, 1958; Fuchs, 1964; Pui et al., 1987). For these calculations, 4.572 mm was taken as the inner diameter of the sampling lines due to a wall thickness of 0.889 mm. For details of the calculations, see Supplemental Material. The transport efficiency for different instruments can vary due to the various sampling line lengths. In general, longer sampling lines lead to more losses of aerosol particles in the coarse mode size range.

### 2.3 Aerosol Number Size Distribution

For this study, the comparison of the in-cabin and out-cabin aerosol number size distribution (NSD) in the coarse mode size range plays a fundamental role in quantifying losses and enhancements of aerosol particles due to the sampling system. Therefore, for each of the 262 A-LIFE flight sequences at a constant altitude, a log normal fitted in-cabin and an out-cabin NSD ($n_{in}$ and $n_{out}$) was derived. Table S2 in the Supplemental Material summarizes the instrumentation used for the derivation of the aerosol NSD inside and outside the aircraft cabin. In this section, a brief explanation of the combined NSD retrieval from measurements of multiple instruments, which we applied for this study, is given. A detailed description of the combined NSD retrieval is given in Dollner (2022).

For the in-cabin NSDs, the measurements of two optical particle counters (OPCs), namely the wing-mounted active sampling Ultra-High Sensitivity Aerosol Spectrometer-Airborne (UHSAS-A) from the manufacturer Droplet Measurement Technologies (DMT; Longmont, CO, USA) and the in-cabin SkyOPC (model 1.129; GRIMM Aerosol Technik; Ainring, Germany) were combined into a dry in-cabin NSD. The dry in-cabin NSD covers the particle diameter size range from 125 nm to about 3 µm. For the out-cabin NSDs, which represents the "full" particle size range, measurements of the UHSAS-A and the SkyOPC at dry conditions were combined with coarse mode size distribution measurements of a wing-mounted DMT Second Generation Cloud, Aerosol, and Precipitation Spectrometer (UNIVIE CAPS; Spanu et al., 2020; Dollner (2022); Dollner et al., 2023a, in prep.) detecting NSDs at ambient relative humidity conditions which were converted to dry particle sizes (for details see below). For the out-cabin NSD, UHSAS-A and SkyOPC measurements were only used up to a particle diameter of 1 µm. For particles in the coarse mode size range, the measurements of the Cloud and Aerosol Spectrometer (UNIVIE CAS) component of the UNIVIE CAPS were used in the particle diameter range between 0.9 µm and 50 µm. The in-cabin aerosol NSD is impacted by losses or enhancements due to the sampling system (inlet + sampling lines), whereas the out-cabin aerosol NSD is largely unaffected by inlet losses. However, as described in Spanu et al. (2020), flow distortions at the location of wing-mounted instruments such as the UNIVIE CAPS can produce artefacts and alter the measured ambient



particle number concentration. The proposed correction scheme in Spanu et al. (2020) was applied to the UNIVIE CAS data to represent the "true" ambient aerosol. The number concentrations for in- and out-cabin NSDs were converted to STP conditions (standard temperature = 273.15 K and standard pressure = 1013.25 hPa), and NSDs refer to dry particle diameters.

265

In this study, we are interested in understanding the efficiency of the Falcon aerosol inlet for coarse mode aerosol measurements. Therefore, the overlapping size range of the measurements of the in-cabin SkyOPC and the wing-mounted UNIVIE CAS (overlapping size range: ~ 0.9 µm - 3 µm) is crucial when comparing the in-cabin with the out-cabin NSD. To determine precise differences between the NSDs, the measurements of the SkyOPC and the UNIVIE CAS were treated in a consistent way, which will be briefly outlined in the next paragraphs. For a detailed description of the calculation of the size distribution, the reader is referred to Walser et al. (2017) for the SkyOPC data and to Dollner et al., in prep. (2023b) for the UNIVIE CAS data, respectively.

270

In order to derive a NSD from OPC measurements, which represents the ambient aerosol mixture accurately, the choice of the aerosol type-dependent refractive index and, hence, the scattering cross-section to particle diameter relationship, is critical (Rosenberg et al., 2012; Walser et al., 2017). For the A-LIFE data analysis, the refractive indices were inferred from the aerosol composition determined with the FLEXPART model (Stohl et al., 1998). The FLEXPART model predicted the mass concentration of five individual aerosol types (mineral dust, black carbon, sea salt, organic matter and sulfate) along the flight track with a temporal resolution of one minute. Additionally, the modelled mass concentrations are given for two diameter size ranges: accumulation mode (100 nm – 1 µm) and coarse mode (1 µm – 10 µm). Black carbon is represented only in the accumulation mode. Based on the contributions of the five aerosol types derived from FLEXPART for each flight sequence in the respective measurement range of the SkyOPC and the UNIVIE CAS, ensembles of refractive indices (based on a literature review for each aerosol type, see Supplemental Material – in particular Table S3) and the corresponding theoretical scattering cross-section to particle diameter relationships were created. The theoretical scattering cross-sections were calculated with Mie-Lorenz theory for the aerosol types black carbon, organic matter, and sulfate by assuming a spherical shape of the particles (Mie, 1908). For mineral dust, prolate spheroids with aspect ratios ranging from 1.2 to 5.0 were assumed and the scattering cross-sections calculated with the T-Matrix method (Waterman, 1965). A cubic shape was assumed for sea salt and the calculations were done with the discrete dipole approximation code ADDA (Yurkin and Hoekstra, 2011). The NSDs were then derived 1000 times for each flight sequence with random draws from this ensemble of scattering cross section functions (Walser et al, 2017; Dollner, 2022). Since the NSDs derived from optical particle counters consider the aerosol composition (refractive index) and the shape of the particles, the indicated particle size refers to geometric diameters. For simplicity, in the following text and plots, only the term particle diameter will be used.

As mentioned previously, the UNIVIE CAS detected aerosol particles at ambient relative humidity conditions, whereas the in-cabin SkyOPC, and the UHSAS-A measured the dry particle size distribution. Depending on the ambient relative humidity,



this could lead to a disagreement between the SkyOPC and the UNIVIE CAS NSD. The FLEXPART-modelled aerosol composition also allowed us to calculate the growth factor of the ambient aerosol particles in order to determine their dry diameter based on Kappa-Köhler theory (Petters and Kreidenweis, 2007; Brock et al., 2016). Representative hygroscopicity parameters for each of the aerosol types (except for black carbon which is assumed as hydrophilic) were taken from literature
and with the resulting growth factors, the out-cabin NSDs were converted to dry particle diameters. An overview of the hygroscopicity parameters as well as the refractive indices, used to calculate the scattering cross-section to particle diameter relationship, is provided in the Supplemental Material (Table S3).

As a last step, the in-cabin and out-cabin NSDs were fitted with three log-normal functions representing the Aitken,
accumulation and coarse mode. The fits were restricted by the integral particle number concentration measured by the TSI 3760a Condensation Particle Counter (CPC2; TSI Inc.; Aachen, Germany) so that total number concentration calculated from the fitted number concentrations in the size range between 10 nm and 50 µm matched the particle number concentration measured with the TSP 3760a CPC. The resulting log-normal fitted dry NSDs ($n_{in}$ representing the in-cabin NSD and $n_{out}$ representing the out-cabin NSD) were used in this study for the characterization of the sampling system.

## 310 2.4 Sampling System Characterization

The characterization of the aerosol sampling system aboard the Falcon aircraft during the A-LIFE field campaign consists of three parts: First, each of the 262 flight sequences was assigned to one of three groups with respect to the sampling conditions: "depletion" ($U_0/U < 1$), "representative sample" ($U_0/U \sim 1$), or "enrichment" ($U_0/U > 1$) of coarse mode aerosol particles in the in-cabin NSD compared to the out-cabin NSD. Second, the sampling efficiencies during measurement periods inside
mineral dust layers ("dust sequences") were derived by comparing the in-cabin to the out-cabin NSD. As stated earlier, the sampling efficiency determined as described is affected by both, inlet and transport losses. Therefore, the inlet efficiency was derived by dividing the experimentally determined sampling efficiency by the transport efficiency calculated based on equations from literature (see Supplemental Material). Third, the cut-off diameters $D_{p,50,sys}$ were derived from the sampling efficiencies as a function of $v_{TAS}$ and calculated for different particle densities. In the next paragraphs, each of the three steps
is described in more detail.

### 2.4.1 Sampling Condition Classification

For the classification of the sampling conditions, the log-normal fitted in-cabin and out-cabin NSDs were integrated in the size range from 1 to 10 µm. The resulting particle number concentrations inside ($N_{in}$) and outside ($N_{out}$) the aircraft cabin were used for the derivation of the classification criteria. If the difference between $N_{in}$ and $N_{out}$ normalized by $N_{out}$ is between -0.1 and
0.1 (i.e. ±10 %), the measured in-cabin aerosol sample during a specific flight sequence is considered as a "representative sample" of the ambient aerosol.





Cases with $[N_{in} - N_{out}]/N_{out} < -0.1$ are designated as "depletion". In contrast, cases with $[N_{in} - N_{out}]/N_{out} > 0.1$ are labelled as "enrichment". The classification criteria are summarized in Table 1. Figure 3 shows examples for each of the three classes at

three different true airspeeds $v_{TAS}$. In the case of particle depletion (Figure 3a) and particle enrichment (Figure 3c), the in- and out-cabin NSDs deviate from each other for particle sizes above ~1 µm. For the "representative sample" (Figure 3b) observed at $v_{TAS} = 144$ m s$^{-1}$, $n_{in}$ and $n_{out}$ are in agreement within the NSD uncertainties.

**2.4.2 Sampling, Transport, and Falcon Inlet Efficiency**

For the derivation of the sampling efficiency $\eta_{sys}$, the experimentally observed NSDs during the 174 mineral dust flight

sequences were grouped into 4 sets according to the $v_{TAS}$ during the flight sequence: $v_{TAS} < 130$ m s$^{-1}$, 130 m s$^{-1}$ ≤ $v_{TAS}$ < 160 m s$^{-1}$, 160 m s$^{-1}$ ≤ $v_{TAS}$ < 190 m s$^{-1}$, and 190 m s$^{-1}$ ≤ $v_{TAS}$. Furthermore, the desired sampling efficiency $\eta_{sys}$ was calculated as a function of particle diameter by dividing $n_{in}$ with $n_{out}$:

$$\eta_{sys}(D_p) = \frac{n_{in}(D_p)}{n_{out}(D_p)} \tag{2}$$


$\eta_{sys}$ are reported as medians, 16$^{th}$, and 84$^{th}$ percentiles from the resulting set of sampling efficiencies for each $v_{TAS}$ range. The resulting particle size-dependent sigmoidal behaviour of the sampling efficiencies were fitted with a Boltzmann sigmoid function (except for the highest $v_{TAS}$ range, where the sampling efficiency curve was only smoothed).

The transport efficiency $\eta_{tr}$ was calculated for each $v_{TAS}$ range for the sampling line configuration of three different in-cabin instruments (SkyOPC, Aurora 4000 Nephelometer, TAP) by using empirical equations from Thomas (1958), Fuchs (1964), and Pui et al. (1987) for sedimentation and bend losses (see Figure 7). The used length, volumetric flow and bend angle of each sampling line piece, leading from the inlet system to the SkyOPC, are summarized in the Supplemental Material (Table S4). For the first four sampling line pieces the mean of the flow range was used. Furthermore, the transport efficiency was

calculated once with an inclination angle of $\theta = 90°$ (horizontal) and once with $\theta = 0°$ (vertical) and was averaged afterwards. As an input for particle parameters for mineral dust, a density of 2.6 g cm$^{-3}$ (Hess et al., 1998) and a shape factor of 1.2 (Kaaden et al., 2008) was assumed. Furthermore, an average in-cabin temperature of 30°C was estimated and used for the calculations. For the ambient pressure, an averaged value for the respective $v_{TAS}$ range was used (796 hPa, 592 hPa, 409 hPa, and 283 hPa).

Following equation (1), the inlet efficiencies $\eta_{in}$ were derived by dividing the experimentally determined sampling efficiencies by the theoretically derived transport efficiency of the SkyOPC. As a result, all three efficiencies of equation (1) are examined for four different $v_{TAS}$ ranges.





**Table 1: Criteria for classifying each flight sequence in relation to the current sampling condition.**

| Sampling Condition Class | $\dfrac{N_{in} - N_{out}}{N_{out}}$ |
|---|---|
| Depletion of coarse mode particles (super-isokinetic sampling) | < -0.1 |
| Representative sample (isokinetic sampling) | -0.1 – 0.1 |
| Enrichment of coarse mode particles (sub-isokinetic sampling) | > 0.1 |


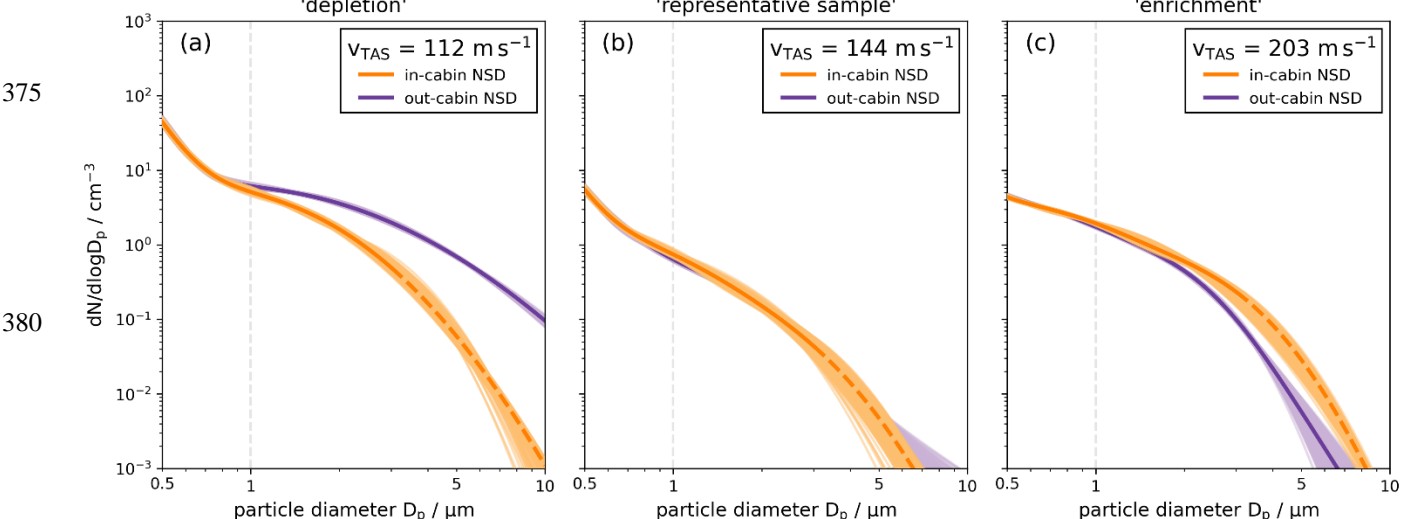

Figure 3: Examples of in-cabin (orange) and out-cabin (violet) aerosol number size distributions (NSDs) during flight sequences with a major contribution of mineral dust particles in the coarse mode size range. For each color, the darker line represents the mean NSDs derived from the aerosol type-dependent refractive index ensemble. For particles larger than 3 µm, the in-cabin mean NSD is shown by dashed lines to indicate that for particles > 3 µm, the in-cabin NSD is only based on the log-normal fits. The lighter-coloured range indicates the NSD uncertainty. Panels (a)-(c) show examples of the three different sampling conditions: (a) Case with depletion of particles larger than 1 µm; (b) Case in which the in-cabin instrumentation is representative of the ambient aerosol; (c) Case of enrichment of coarse mode particles due to sampling effects caused by the inlet system. Note, the corresponding $v_{TAS}$ values are different for all three sampling condition cases. We want to emphasize that the ratio of the velocities $U_0/U$ is decisive whether depletion, representative sampling, or enrichment occurs.



### 2.4.3 Cut-Off Diameter

When an aerosol particle enters a sampling system, it encounters different flows in the inlet and tubing system. Depending on the inertia of the particle, defined by its size and density, it can either adapt to the new flow conditions and be transported to the measurement device or not follow the flows in the inlet and sampling lines and be separated. Whether or not a particle is deposited in the measuring system, can be determined by the Stokes number Stk which is defined by the ratio between the stopping distance S of the particle and a characteristic length L (Hinds, 1999):

$$Stk = \frac{S}{L} = \frac{\rho \, D_p^2 \, C_c \, v_0}{18 \, \eta \, \chi \, L} \tag{3}$$

Here, $\rho$ is the particle density, $D_p$ is the particle diameter, $C_C$ the Cunningham slip correction factor (see Seinfeld and Pandis, 2016), $v_0$ is the velocity of the particle, $\eta$ is the dynamic viscosity of the ambient air (see Seinfeld and Pandis, 2016), and $\chi$ the dynamic shape factor.

The size at which 50 % of the aerosol particles pass the inlet and tubing system is called the cut-off diameter of the sampling system $D_{p,50}$. The concept of cut-off diameters is often used to describe limitations of instruments (e.g. the lower cut-off diameter of an instrument) or sampling systems regarding the detection efficiency. Here, we derive the cut-off diameters of the sampling system $D_{p,50,sys}$ aboard the Falcon during the A-LIFE project for particles with different densities by using the Stokes number Stk. A similar approach was executed in Fiebig (2001) and Porter et al. (1992). In a first step, the cut-off diameters, i.e. the particle size at which the sampling efficiency $\eta_{sys}$ equals 50 %, were determined experimentally from the previously calculated sampling efficiencies for 174 dust sequences. The results were taken to calculate the corresponding Stokes number $Stk_{50,sys}$ with equation (3). For the calculations, we assumed a particle density of 2.6 g cm$^{-3}$ (Hess et al., 1998), and a dynamic shape factor $\chi$ of 1.2 (Kaaden et al., 2008) to represent mineral dust particles. For the aerosol particle velocity $v_0$, $v_{TAS}$ was divided by the diffuser cross section ratio of 7.1 of the Falcon aerosol inlet. The inner diameter of the Falcon isokinetic inlet and the sampling lines (L = 4.572 mm) were used as the characteristic length. The resulting values were fitted with a logarithmic function of $v_{TAS}$ for the range between 70 m s$^{-1}$ and 220 m s$^{-1}$. Supporting the idea that $Stk_{50,sys}$ remains constant for each $v_{TAS}$ value, we converted each $Stk_{50,sys}$ back to a new cut-off diameter $D_{p,50,sys}$ by using averaged values of ambient pressure and temperature and a certain particle density. This method allowed us to derive the cut-off diameter (assuming a Stokes' equivalent sphere with $\chi$ = 1; Hinds, 1999) of the Falcon sampling system for three different particle densities (1.0, 1.8, and 2.0 g cm$^{-3}$). Additionally, we also derived the cut-off diameters for particles with a density of 2.6 g cm$^{-3}$ and shape factor $\chi$ = 1.2 representing mineral dust particles. The fitted ambient pressure and temperature, which were used for these calculations, are displayed in the Supplemental Material (Figure S2 and Figure S3).



## 3 Results

### 3.1 Sampling Condition Classification of the A-LIFE Data Set

As outlined in section 2.4.1, each A-LIFE sequence on constant altitude was classified regarding the present flight sampling conditions. Figure 4a presents the mean $v_{TAS}$ and mean altitude of each flight sequence, with the flight sequence classification

("depletion", "representative sample", "enrichment") highlighted by the colour coding of the markers (with triangles indicating sequences without, and spheres marking sequences with a major contribution of mineral dust in the coarse mode particle size range). During the majority of the A-LIFE flight sequences (167 of 262 cases), a depletion of coarse mode aerosol particles was observed with the in-cabin instrumentation. These cases were sampled at an average altitude of $3.74 \pm 2.40$ km (mean and standard deviation) and at an average $v_{TAS}$ of $136 \pm 23$ m s$^{-1}$. A "representative" sample was measured during 35 sequences.

These sequences occurred at a mean altitude of $4.80 \pm 3.43$ km respectively a $v_{TAS}$ of $146 \pm 35$ m s$^{-1}$. The majority of the sequences with an enrichment of coarse mode particles was observed at altitudes higher than 7 km and $v_{TAS}$ faster than 170 m s$^{-1}$. In total, 60 flight sequences were classified as "enrichment cases". The average altitude of those sequences is $7.80 \pm 2.93$ km and the average $v_{TAS}$ is $177 \pm 31$ m s$^{-1}$. Figure 4b is restricted to the 174 dust sequences which are a subset of all measurements. The colour coding indicates the strength of the respective sampling effects in terms of particle number

concentration in the size range from 1 to 10 µm as defined in Table 1 (given in percentages). As can be seen by Figure 4b, the most pronounced depletion occurs for the lowest $v_{TAS}$. The particle losses in the sampling system decrease with increasing $v_{TAS}$ until the isokinetic sampling range (~ 164 m s$^{-1}$) is reached. For higher $v_{TAS}$, particle enrichment occurs.

### 3.2 Cut-Off Diameter of the Sampling System aboard the Falcon Research Aircraft

Figure 5a depicts the experimentally determined cut-off diameters $D_{p,50,sys}$ of the A-LIFE sampling system (inlet + sampling

lines of the transport system) as a function of $v_{TAS}$ for all 174 A-LIFE mineral dust sequences. The cut-off diameters $D_{p,50,sys}$ span a large range from about 1.1 µm to 9.2 µm. In general, for increasing $v_{TAS}$ values, $D_{p,50,sys}$ also increases. Note, these cut-off diameters $D_{p,50,sys}$ are also affected by the length and geometry (e.g. number of bends, horizontal or vertical tubing lines, etc.) of the sampling lines connecting the SkyOPC with the Falcon aerosol inlet. A difference in the sampling line length and geometry of the aerosol transport system would also change the cut-off diameter $D_{p,50,sys}$ of the entire sampling system (see

Figure 7 for the transport efficiency of the Nephelometer and TAP). For example, the transport efficiency of the Nephelometer shows a cut-off diameter $D_{p,50,tr}$ which is about 26 % smaller than $D_{p,50,tr}$ for the SkyOPC due to longer sampling lines.

Figure 5b illustrates the corresponding Stokes number for 50% sampling efficiency $Stk_{50,sys}$ as a function of $v_{TAS}$ for all 174 A-LIFE dust sequences. The median $Stk_{50,sys}$ for $v_{TAS}$ below 160 m s$^{-1}$ is 0.20 while for larger $v_{TAS}$ values it is 1.41, which

deviates by one order of magnitude. Consequently, a logarithmic function was chosen to fit the $Stk_{50,sys}$ values as a function of $v_{TAS}$ between 70 m s$^{-1}$ and 220 m s$^{-1}$.





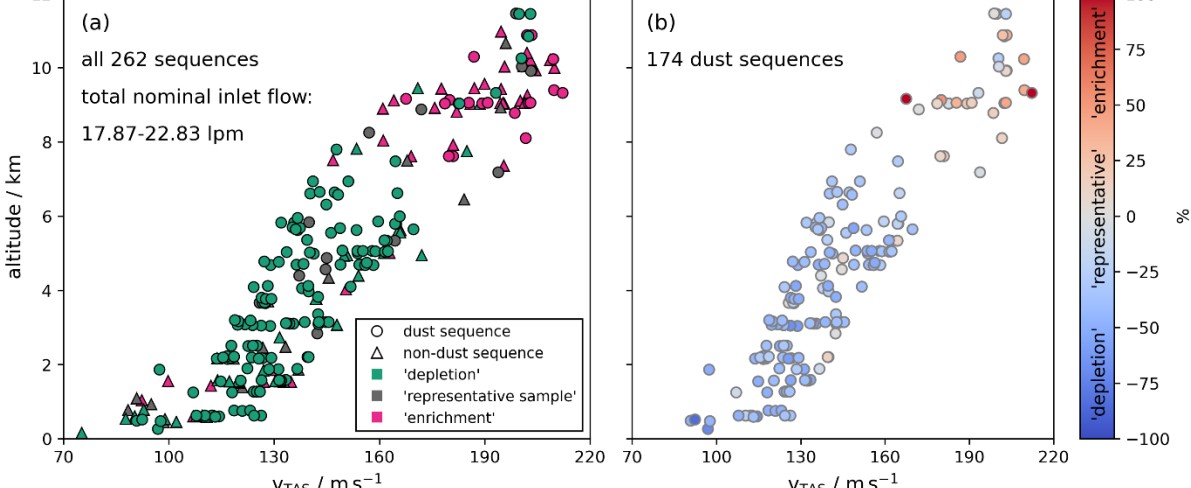



**Figure 4: Vertical profile of mean $v_{TAS}$ as a function of mean altitude for (a) all 262 A-LIFE flight sequences, and (b) the A-LIFE mineral dust sequences. (a) The shape of the markers indicates the presence of mineral dust during the sequence, and the colour represents the sampling condition including "depletion" ($U_0/U < 1$), "representative sample" ($U_0/U \sim 1$), and "enrichment" ($U_0/U > 1$) Panel (b) is restricted to the 174 A-LIFE dust sequences. The colour coding illustrates the strength of the deviation of the in-cabin and the out-cabin particle number concentration in the size range from 1 to 10 μm (given in percentages; see also Table 1). In both panels, it is visible that enrichment cases occur at the higher end of observed $v_{TAS}$ values.**






**Figure 5: (a) Experimentally derived cut-off diameters $D_{p,50,sys}$ of the Falcon aerosol sampling system (inlet + tubing lines) for the 174 A-LIFE mineral dust sequences as a function of true airspeed $v_{TAS}$. (b) Corresponding Stokes number for 50 % sampling efficiency as a function of $v_{TAS}$. The red line represents logarithmic fit of $Stk_{50,sys}$ as a function of $v_{TAS}$. The red-shaded area indicates the fit uncertainty.**



The cut-off diameter of the Falcon aerosol sampling system $D_{p,50,sys}$ was also assessed for different aerosol types characterized by different particle densities. This was achieved by converting the fitted $Stk_{50}$ values for the $v_{TAS}$ range from 70 to 220 m s$^{-1}$ to cut-off diameters $D_{p,50,sys}$ under the assumption of different particle densities. Figure 6 shows the results for four different particle densities. Although, a $v_{TAS}$ can vary substantially at a given flight altitude as shown in Figure 2, a corresponding mean

flight altitude for $v_{TAS}$ ranging between 120 and 180 m s$^{-1}$ is included on the right side of the ordinate of Figure 6. Similar to the experimentally determined cut-off diameters $D_{p,50,sys}$ in Figure 5a, the derived cut-off diameters also increase with an increasing $v_{TAS}$. As expected, a larger cut-off diameter is observed for smaller particle densities. For unit density, the cut-off diameters range from 3.6 µm at a flight altitude of 2 km to 6.4 µm at 8 km. Doubling of the particle density (2 g cm$^{-3}$), decreases the cut off-diameter $D_{p,50,sys}$ to 2.5 µm at 2 km and to 4.5 µm at 8 km.

**3.3 Sampling System Efficiencies for Different Airspeed Ranges**

In section 3.2, the experimentally determined cut-off diameters of the A-LIFE sampling system $D_{p,50,sys}$ were presented as a function of $v_{TAS}$, and for different particle densities. In this section, the sampling efficiency of the A-LIFE aerosol sampling system is presented for different airspeed ranges as a function of particle diameter. Furthermore, the inlet efficiency is derived as a function of particle diameter by dividing the experimentally derived sampling efficiencies by the theoretically derived

transport efficiency of the SkyOPC. For all cases with particle depletion, the particle size with 50 % efficiency was determined for all three efficiencies. An overview of the determined cut-off diameters of the sampling system ($D_{p,50,sys}$), inlet system ($D_{p,50,in}$), and transport system ($D_{p,50,tr}$) for different airspeed ranges during A-LIFE dust sequences is given in Table 2.

Figure 7 shows the theoretically-calculated transport efficiency $\eta_{tr}$ for various measurement devices including the SkyOPC,

an Aurora 4000 Nephelometer, and a Tri-color Absorption Photometer (TAP) over an ambient pressure range from 283 to 796 hPa (which are the average values for the first and last $v_{TAS}$ range). The transport efficiency of the SkyOPC is depicted in violet. The violet line shows the case of mineral dust, which is used in this study to derive the inlet efficiency during dust sequences. For comparison, the violet-black striped line depicts the corresponding transport efficiency for particles with unit density and unit shape factor. The upper cut-off diameter (i.e. the 50 % cut-off diameter at the larger end of the particle size distribution) for the A-LIFE transport system $D_{p,50,tr}$ inside the aircraft cabin for the SkyOPC ranges from 3.6 µm at 283 hPa

to 3.8 µm at 796 hPa for mineral dust particles which suggests that a changing ambient pressure has only a minor influence on the transport efficiency for coarse mode aerosol particles in the A-LIFE transport system inside the aircraft cabin. The $D_{p,50,tr}$ increases to 5.4 µm at 283 hPa to 5.6 µm at 796 hPa for particles with unit density and unit shape factor. For comparison, Figure 7 also shows the theoretically-calculated transport efficiency for the Nephelometer and the TAP instruments which

show upper cut-off diameters $D_{p,50,tr}$ that are 25-27 % and 26-28 % smaller compared to the SkyOPC due to the longer sampling lines (see Table S1 in the Supplemental Material).














**Figure 6:** Cut-off diameter of the sampling system as a function of $v_{TAS}$ for four different particle densities (Panel (a)-(d)) derived from the logarithmic fit of the Stokes number shown in Figure 5b. On the right y-axes a mean altitude to the corresponding $v_{TAS}$ ranging from roughly 2 to 8 km is shown. In panel (a) and panel (d) the experimentally derived cut-off diameters from Fiebig (2001) are included. Panel (d) also shows the derived cut-off diameters $D_{p,50,sys}$ from the NSDs in Figure 5a.










**Figure 7: Transport efficiency for three different in-cabin instruments deployed during A-LIFE focussing on the coarse mode size range. For all calculations mineral dust particles were assumed with a particle density of 2.6 g cm⁻³ and a shape factor of 1.2. For the SkyOPC, the transport efficiency was also calculated for particles with unit density and unit shape factor (violet-black striped line). The computations cover the ambient pressure range from 283 to 796 hPa (which are the average values for the first and the last $v_{TAS}$ range as defined in section 2.4.2) represented by the width of the lines, and assume an in-cabin temperature of 30°C.**





Figure 8 shows the measured sampling efficiency $\eta_{sys}$ (left column, blue lines), the calculated transport efficiency $\eta_{tr}$ of the SkyOPC (right column, violet lines), and the inferred Falcon aerosol inlet efficiency $\eta_{in}$ (right column, green lines) as a function of particle diameter $D_p$ during dust sequences. Each row represents a specific $v_{TAS}$ range. The sampling efficiencies were derived from the log-normal fitted NSDs extending to 10 µm diameter whereas the inlet efficiencies are only given for

the measurement size range of the SkyOPC up to 3 µm.

In general, for increasing $v_{TAS}$ and therefore increasing altitude, an increasing cut-off diameter of the sampling system $D_{p,50,sys}$ is observed. While the $D_{p,50,sys}$ only increases slightly from the first to the second $v_{TAS}$ range (from 2.4 µm in Figure 8a to 2.6 µm in Figure 8b), the differences are larger for the last two $v_{TAS}$ classes ($D_{p,50,sys}$ increased from 4.5 µm in Figure 8c to 5.9 µm in Figure 8d). Up to a $v_{TAS}$ of 190 m s$^{-1}$, the averaged sampling efficiencies stay below 100 % along the entire particle diameter

range. Only the last $v_{TAS}$ range (Figure 8d) shows values that exceed 100 % which can be associated with the enrichment of coarse mode particles due to the Falcon aerosol inlet. For $v_{TAS}$ higher than 190 m s$^{-1}$, the sampling efficiency reaches a maximum of 122 % at a particle diameter of 2.7 µm. For larger particles, the sampling efficiency shows a steep decrease with values of less than 100 % for particle diameters larger than 4.0 µm.

The inlet efficiencies $\eta_{in}$ (green lines in Figure 8e-h) show the following behaviour: In the first two $v_{TAS}$ ranges below 160 m s$^{-1}$ (Figure 8e and Figure 8f), the inlet efficiencies indicate a depletion of coarse mode particles in the inlet system. In the first $v_{TAS}$ range, the cut-off diameter of the inlet system $D_{p,50,in}$ is about 2.9 µm. In the second $v_{TAS}$ range, the inlet efficiency didn't drop below 50 % suggesting that the cut-off diameter $D_{p,50,in}$ has to be larger than 3 µm. Between a $v_{TAS}$ of 160 and 190 m s$^{-1}$ (Figure 8g), the data reveal a nearly unit inlet efficiency with a slight tendency of enrichment starting at particle diameters of

approximately 2.0 µm. For $v_{TAS} > 190$ m s$^{-1}$ (Figure 8h), coarse mode particles are enriched in the inlet system with an enrichment factor of 1.31 at a particle diameter of 2.0 µm which increases to 1.76 at 3.0 µm particle diameter.

## 4 Discussion

### 4.1 Characterization of the Falcon Sampling System

From the characterization of the sampling system of the Falcon research aircraft two major findings emerged: First, the sampling efficiency is highly dependent on the true airspeed of the aircraft with respect to the surrounding air mass. During A-LIFE we encountered all three different sampling conditions (sub-isokinetic, isokinetic, and super-isokinetic) which have different influences on the sampling of coarse mode aerosol particles with in-cabin instruments. A key step for characterizing the Falcon aerosol sampling system was to group the flight sequences according to the $v_{TAS}$ instead of the flight altitude since

the velocities in and outside the sampling system are the physical parameters that govern the sampling conditions.







**Figure 8: Sampling efficiency (left panels) and inlet and transport efficiency (right panels) as a function of particle diameter $D_p$. Each row represents a specific $v_{TAS}$ range. The dark blue line of the sampling efficiency indicates the median of all calculations while the light blue area shows the uncertainty range between the 16th and 84th percentile. The solid blue line indicates the measurement range of the SkyOPC. Above ~3 µm in diameter (dashed blue line), data from the in-cabin NSD fit were used to derive the sampling efficiency. The transport efficiency shows theoretical calculations for the transport system of the SkyOPC. The inlet efficiency was derived from the experimental approach for the sampling efficiency and the theoretical calculations for the transport efficiency and is shown here up to a particle diameter of 3 µm.**



**Table 2: Overview of the determined cut-off diameters (A-LIFE transport System, Falcon aerosol inlet, and sampling system) for different $v_{TAS}$ ranges during the A-LIFE aircraft field campaign. The experimentally determined cut-off diameters for the sampling system, $D_{p,50,sys}$, were derived from measurements in mineral dust sequences (particle density about 2.6 g cm$^{-3}$). The cut-off diameters for the transport system, $D_{p,50,tr}$, were theoretically calculated assuming also a particle density of 2.6 g cm$^{-3}$, and a particle shape factor of 1.2. Consequently, the cut-off diameter of the Falcon aerosol inlet, $D_{p,50,in}$, also refers to a particle density of 2.6 g cm$^{-3}$.**

| $v_{TAS}$ range | Cut-off diameter transport system (i.e. effect of the sampling lines) $D_{p,50,tr}$ | Cut-off diameter Falcon aerosol inlet (i.e. effect of the inlet only) $D_{p,50,in}$ | Cut-off diameter sampling system (i.e. combined effect of inlet and sampling lines) $D_{p,50,sys}$ |
|---|---|---|---|
| $v_{TAS} < 130$ m s$^{-1}$ | 3.8 µm | 2.9 µm | 2.4 µm |
| 130 m s$^{-1} \leq v_{TAS} < 160$ m s$^{-1}$ | 3.7 µm | > 3 µm | 2.6 µm |
| 160 m s$^{-1} \leq v_{TAS} < 190$ m s$^{-1}$ | 3.7 µm | Isokinetic to sub-isokinetic sampling conditions | 4.4 µm |
| 190 m s$^{-1} \leq v_{TAS}$ | 3.6 µm | Sub-isokinetic sampling conditions | 5.8 µm |



Second, the results of this study show the significance that sampling efficiencies always have to be considered as a combination of both, the inlet and the transport efficiency. The combination of the two efficiencies is especially evident in the sampling efficiency of the largest $v_{TAS}$ range (190 m s$^{-1}$ ≤ $v_{TAS}$; Figure 8d and Figure 8h). Theoretical estimations of the inlet efficiency at sub-isokinetic sampling conditions suggest an increasing enhancement for increasing particle size. However, Figure 8 shows that an enhancement of particles only occurs up to a particle diameter of around 4 µm (with a peak at 2.7 µm). For larger

particles, the losses through the sampling lines are stronger than the enhancement due to the inlet, which leads to a rapidly dropping sampling efficiency of the entire aerosol sampling system. This underlines the important role of the transport efficiency in the overall detection efficiency of aerosol particles.

As mentioned in section 1, Fiebig (2001) calculated the cut-off diameters of the sampling system aboard the Falcon as a

function of flight altitude. For the isokinetic Falcon aerosol inlet, 6 cut-off diameters and the corresponding Stokes number were experimentally derived from instrument data collected within the LACE 98 experiment within a $v_{TAS}$ range between 104 and 118 m/s under super-isokinetic sampling conditions (resulting in a depletion of coarse mode particles at the inlet system). These results were taken to calculate one mean value of the Stokes number $Stk_{50,Fiebig}$ = 0.486 ± 0.137 which was used to convert the cut-off diameters for the entire altitude range of the Falcon assuming altitude-dependent average values for ambient

temperature, pressure and airspeed. The main aerosol types and therefore the particle density during the investigated flight sequences, used to derive the cut-off diameters, were not specified in Fiebig (2001). Therefore, we included the data points from Fiebig (2001) in Figure 6a and Figure 6d for comparison thereby covering the range of particle densities between 1.0 g cm$^{-3}$ and 2.6 g cm$^{-3}$. The smaller cut-off diameters from Fiebig (2001) match with our derived cut-off diameters for a particle density of 2.6 g cm$^{-3}$ (see Figure 6d) while the larger cut-off diameters are in agreement with our derived cut-off diameters for

unit particle density (see Figure 6a). The cut-off diameters from Fiebig (2001) are consistent with our results for particle densities between 1.0 and 2.6 g cm$^{-3}$ despite several differences in the set-up. For example, the total flow velocity of the in-cabin instrumentation during the LACE 98 aircraft project was with a total flow of 27.3 m s$^{-1}$ much larger than during A-LIFE where the total flow velocity was 15 % to 33 % smaller. Furthermore, the geometry of the transport system was very likely different from the A-LIFE in-cabin set up. Although, the experimentally-determined $D_{p,50,sys}$ in this study agree with what is

referred-to as "cut-off diameters" in Fiebig (2001), the altitude-dependent cut-off diameters inferred from the Stokes number are different to this study: in Fiebig (2001), the cut-off diameter decreases (for particles with unit density) from 2.3 µm at ground level to 1.7 µm at around 8 km altitude as a result of using the same mean Stokes number for the entire altitude range. As visible from equation (3), keeping the Stokes number constant, but increasing the altitude leads to an increase in the true airspeed of the aircraft which causes decreasing cut-off diameters with altitude. We believe that continuous values of the Stokes

number over the entire airspeed range are needed to cover all three sampling conditions. The A-LIFE data set enabled us to determine the cut-off diameters during three different sampling conditions (sub-isokinetic, isokinetic, and super-isokinetic) and the results show an increasing value of $D_{p,50,sys}$ for an increasing altitude and airspeed, respectively (see Figure 6).



## 4.2 Uncertainties and Limitations of the Study

Every study is subject to uncertainties and limitations. In the following paragraphs we discuss different aspects that have to be kept in mind when referring to the results of this study.

*Fixed total volumetric flow.* As mentioned in section 1, the ratio between the ambient air velocity and the stream velocity inside the inlet plays a crucial role in enabling the measurement of a representative sample of aerosol particles. During the A-

LIFE campaign, the in-cabin instrumentation was always fully operated, which means that the stream velocity inside the inlet was always in the range between 18.14 and 23.18 m s$^{-1}$. During A-LIFE as well as in previous studies, the in-cabin volume air flow in the Falcon aerosol sampling system was not controlled (e.g. Minikin et al., 2003; Petzold et al., 2009; Schumann et al., 2011; Weinzierl et al., 2017; Moore et al., 2017). Therefore, isokinetic flow conditions were only matched at certain flight altitudes. Future measurement campaigns with the Falcon could have a different total volumetric flow (e.g. due to another set

of in-cabin instruments) which could lead to sampling conditions occurring at other airspeeds compared to this study. According to the $U_0/U$ ratio, with a lower total volume flow the enhancement effect is expected to already happen at lower airspeeds and therefore lower flight altitudes.

*Aerosol inlet system of the Falcon.* The theory behind the possible depletion or enhancement of coarse mode aerosol particles

is, strictly speaking, only valid for the inlet opening of a sampling tube and can be described with the particle size-dependent inlet efficiency $\eta_{in}$ (see Hinds, 1999 or Brockmann, 2011). Here, we assign the inlet efficiency to the whole inlet system shown in Figure 1 that transports the ambient aerosol particles from the environment to the transport system components inside the aircraft. This includes the diffusor, the tube with the 90° bend, and the isokinetic inlet tube (see Figure 1). However, this inlet system is used for every flight with the Falcon and does not change during different measurement campaigns. Therefore, it

might be plausible to describe this inlet system with one particle size-dependent transmission efficiency $\eta_{in}$. It is also worth mentioning that nearly isokinetic sampling conditions are present in the third $v_{TAS}$ range (160 m s$^{-1}$ $v_{TAS}$ > 190 m s$^{-1}$; Figure 8g), where the inlet efficiency shows about 100 % up to a particle size of 2 µm, while such conditions are predicted (see section 2.2) in a lower $v_{TAS}$ range between 129 and 165 m s$^{-1}$. A possible explanation for this might be that we assign only one inlet efficiency to the whole inlet system as just described above.


*Combination of experimentally derived efficiencies with theoretical calculations.* In order to estimate the inlet efficiency, the sampling efficiency and the transport efficiency has to be known (see equation (1)). In this study, the transport efficiency was derived with empirical equations from literature in the particle diameter range from 800 nm to 10 µm. The resulting transport efficiencies show an exponential decrease for particles larger than ~2-3 µm (Figure 7). The experimentally derived sampling

efficiencies were determined from the log-normal parametrized NSD. For the in-cabin NSD, the data from the SkyOPC were used which has an upper size detection limit of about 3 µm in the setting used during A-LIFE. Beyond 3 µm, the fit of the in-



cabin NSD was used up to 10 µm diameter (see also Figure 3). Although the fits were restricted so that total number concentration calculated from the fitted number concentrations in the size range between 10 nm and 50 µm matched the integral particle number concentration measured with a CPC, the sampling efficiencies for particles larger than 3 µm are based on the

log-normal fits (blue dashed line in left column of Figure 8). By comparing the sampling and transport efficiency one can see that the sampling efficiency does not decrease as steep as the theoretical transport efficiencies which leads to inlet efficiencies with higher uncertainties. Due to this reason the inlet efficiencies are only shown up to a particle size of 3 µm in Figure 8.

*Coarse mode aerosol composition.* For the derivation of the sampling efficiencies we focussed only on the dust cases, based

on the aerosol classification for the A-LIFE campaign (Weinzierl et al., 2023, in prep.). According to the FLEXPART model, these cases provided a major contribution of mineral dust particles in the coarse mode size range from 1 to 10 µm in diameter. The dust contribution during these flight sequences accumulated to 91 % among all aerosol types, when averaging the FLEXPART data. Hence, this approach was chosen to enable the derivation of the Stokes number with a better-constrained choice of particle density and dynamic shape factor. But, the sampling efficiencies shown in this present work can deviate

when the major contributor in the coarse mode size range are not mineral dust particles and the particles that have a different density and subsequently also inertia (e.g. sea salt or sulfate which have a lower density than mineral dust; see Hess et al., 1998).

## 5 Conclusion

Measurements of ambient aerosol particles inside the cabin of a fast-flying research aircraft is a challenging task due to high

airspeeds and to changing ambient conditions. In particular, for high-quality coarse mode aerosol measurements, understanding of particle losses and enhancements in the aerosol sampling system (formed by the inlet and the transport system) is crucial. The aim of this study was to characterize the aerosol sampling system aboard the Falcon research aircraft. The characterization was carried out using data from the A-LIFE aircraft field experiment in April 2017. The A-LIFE data set was divided into 262 flight sequences on constant altitude with homogenous aerosol conditions outside of clouds. In-cabin and out-cabin particle

number size distributions were calculated for each flight sequence and were key inputs for the characterization of the Falcon aerosol sampling system. Each flight sequence on constant altitude was assigned to one of three groups "depletion" ($U_0/U <$ 1), "representative sample" ($U_0/U \sim 1$), or "enrichment" ($U_0/U > 1$) based on true airspeed of the research aircraft, and particle size-dependent sampling efficiencies were calculated for different true airspeed ranges. The cut-off diameter (50 % sampling efficiency) of the sampling system $D_{p,50,sys}$ was also determined as a function of true airspeed for each constant altitude flight

sequence with a major contribution of mineral dust particles. Based on the experimentally derived Stokes numbers, the cut-off diameters $D_{p,50,sys}$ were calculated for different particle densities as a function of true airspeed.



During the A-LIFE field experiment, the Falcon research aircraft flew with true airspeeds (in this study described as $v_{TAS}$) between roughly 70 and 220 m s$^{-1}$ which means that all three sampling conditions (sub-isokinetic, isokinetic, and super-isokinetic) were encountered during the project. As described earlier, the Falcon aerosol inlet uses a diffusor slowing down the air by a factor of 7.1, and the in-cabin aerosol instrumentation operated during A-LIFE drew a total volume flow which ranged between 18.14-23.18 m s$^{-1}$ at the isokinetic inlet's entry. A depletion of coarse mode particles due to the sampling system was observed during most of the flight sequences at the lowest mean $v_{TAS}$ of about $136 \pm 23$ m s$^{-1}$. Representative samples of ambient aerosol particles were detected by the in-cabin instrumentation (referring to isokinetic sampling conditions) at a mean $v_{TAS}$ of $146 \pm 35$ m s$^{-1}$. Enrichment of coarse mode particles happened at a $v_{TAS}$ of $177 \pm 31$ m s$^{-1}$. The results of the sampling condition classifications show that the velocity of the research aircraft has a major impact on the sampling efficiency of coarse mode aerosol particles with in-cabin instruments. Contrary to past studies, where often the flight altitude was used to characterize the sampling system, we recommend using the airspeed of the aircraft with respect to the surrounding air mass (true airspeed) for the characterization since $v_{TAS}$ together with the flow velocity drawn by the aerosol instrumentation determines the $U_0/U$ ratio (see section 1).

The measured particle size-dependent sampling efficiencies during dust sequences show that up to $v_{TAS}$ of 190 m s$^{-1}$, in most cases, only losses of particles occur in the sampling system. For higher $v_{TAS}$, a size dependent enhancement effect was observed up to a particle diameter of 4.0 µm with a maximum at 2.6 µm. Even larger particles are still enhanced in the inlet system, but inertial and gravitational particle losses in the transport system get more and more pronounced which leads to a decreasing overall efficiency of the aerosol sampling system. This demonstrates the importance of considering both, inlet and transport efficiency, when quantifying the overall sampling efficiency of an aerosol inlet system.

The cut-off diameters $D_{p,50,sys}$ were derived from the particle size-dependent sampling efficiencies. The results show that the cut-off diameters increase with an increasing $v_{TAS}$ which is in accordance with the previously demonstrated presence of an enhancement effect of coarse mode particles at high $v_{TAS}$. By using the corresponding Stokes number $Stk_{50,sys}$ as a function of $v_{TAS}$, the cut-off diameters were calculated for four different particle densities (1.0, 1.8, 2.0, and 2.6 g cm$^{-3}$). For a higher particle density, lower cut-off diameters result over the entire $v_{TAS}$ range.

In an ideal aerosol sampling system on a research aircraft, the volumetric flow of the in-cabin instrumentation should match the estimated (and by the diffuser decelerated) airspeed at all flight altitudes. This may be achieved by a controllable flow that everywhere matches the actual airspeed to always allow near isokinetic sampling conditions ($U_0/U \sim 1$), e.g. through adding a controllable by-pass flow. In cases when the in-cabin instrumentation already needs a high flow, this may require adding an additional aerosol inlet. Furthermore, sampling lines should be designed to be as short as possible and to contain as few bends as possible to avoid losses. Instruments measuring at both ends of the size distribution should be installed as close as possible to the inlet to minimize diffusional (small particles), as well as gravitational and inertial losses (large particles). However,



external factors (e.g. airworthiness certification restrictions, limitations in the number of aerosol inlets, limited space for instruments near the inlet etc.) often constrain the sampling setup so that it is not always possible to deploy a near-ideal aerosol sampling system. In any case, it is important to know the limitations of the aerosol sampling system consisting of both, the

inlet and the transport system.

*Data availability.* The fit parameters of the log-normal number size distributions of the 262 flight sequences and a list of the sampling condition as well as mean $v_{TAS}$, pressure, temperature and altitude during all flight sequences will made publicly available in the University of Vienna data archive Phaidra (https://doi.org/10.25365/phaidra.368).


*Author contributions.* MS presented the idea, analyzed the data and wrote the manuscript with the help of BW. BW coordinated the A-LIFE project. BW and MD performed the aerosol measurements during the A-LIFE field campaign. AT and PS performed the aerosol analysis with FLEXPART simulations. MS, MD, and JG calculated the combined number size distribution. All co-authors participated in the scientific discussion and reviewed the manuscript.


*Competing interests.* The authors declare that they have no conflict of interest.

*Financial support.* This project has received funding from the European Research Council (ERC) under the European Union's Horizon 2020 research and innovation framework program under grant agreement No. 640458 (A-LIFE), and the ESA project
A-CARE (ESA Contract No. 4000125810/18/NL/CT/gp). In addition, the German Aerospace Center (DLR) provided funding for a significant amount of flight hours and aircraft allocation days for the A-LIFE aircraft field experiment. Two EUFAR projects were also clustered with A-LIFE and provided funding for 16 flight hours. Furthermore, MS received financial support from the Vienna Doctoral School in Physics (VDSP). Open access funding provided by the University of Vienna.

*Acknowledgements.* We thank the A-LIFE team and our local hosts in Cyprus for their support and the great collaboration. We are grateful to the DLR Institute of Atmospheric Physics and the DLR Flight Experiments, in particular to Andrea Hausold, the pilots, and the technical and sensor team from DLR flight operations for their great support. Furthermore, we would like to thank Heidi Huntrieser (DLR) and Robert Wagner (TROPOS) for preparing the A-LIFE weather forecasts and for their help with flight planning We thank ZAMG (now Geosphere) for providing access to ECMWF forecast data to calculate the
trajectories in real time, and to the CAMS User Support team. Copernicus Atmosphere Monitoring Service (CAMS) information, partly modified, was used for this paper; neither the European Commission nor ECMWF is responsible for any use that may be made of the information.

*Disclaimer.* The views expressed in this study are those of the authors and do not necessarily represent the views of the CTBTO
Preparatory Commission



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
