# Peer review of "Characterization of the airborne aerosol inlet and transport system used during the A-LIFE aircraft field experiment"

_EGUsphere, 2023_

## Author Response (AR1)

**We want to thank Referee #1 for providing valuable comments on our manuscript which helped to improve our manuscript. Our replies are given in bold.**

This study investigates the sampling efficiency of an aerosol inlet system implemented on the DLR Falcon using a combination of observations and models to quantify cut-off diameters as well as particle losses and enhancements. The manuscript is well-written and provides useful information to help guide those researchers who use measurements from this system to study atmospheric aerosols and their impact on weather and climate. There are a number of questions that I have that prevent me from immediately accepting this paper for publication. Most of my questions concern components of the study that are missing, without which the manuscript remains incomplete. Others are more about clarification of key points.

1.) At various points in the manuscript, as well as in the figures, uncertainties in the derived efficiencies are mentioned in the text or drawn on the figures; however, I didn't find a formal uncertainty analysis and error propagation that quantifies these uncertainties, either in the main or supplemental material. This type of analysis is mandatory for a study like this, how else to interpret the large standard deviations in Figure 8, left column? The figure caption states that these are uncertainties, but unless these are drawn from the uncertainty analysis, then this figure is mis-labeled. Standard deviations about the mean are not uncertainties but actually natural fluctuations in the measurement. Please add a section that goes step by step through where every source of uncertainty is and then propagate those uncertainties (errors) using root sum square (RSS) analysis.

**For each of the 262 flight sequences at constant altitude during the A-LIFE campaign, a log normal fitted aerosol number size distribution (NSD) was derived and applied in this paper to study the sampling system aboard the Falcon research aircraft. For the calculation of the NSDs, a Monte Carlo method was used to cover aerosol specific (e.g. scattering cross-section to particle diameter relationship) as well as instrumental (e.g. calibration parameters, spectral broadening as described in Walser et al., 2017) uncertainties. Therefore, for each flight sequence 1000 NSDs were derived with randomly selected input parameters. These NSDs were used in the derivation of the sampling efficiency (depicted in Figure 8 in the manuscript). Figure 8 shows the median sampling efficiency as well as the 16th and 84th percentiles from all of the calculated NSDs in the respective $v_{TAS}$ range.**

**For clarification, we added more information on this in section 2.3 and 2.4.2 in the main manuscript and under the 3rd paragraph of this reply.**

2.) Three key aspects of the inlet system that are not even mentioned are i) the angle of flow into the inlet, ii) turbulence in the inlet and iii) compressional heating. This is a blunt inlet, not a sharp edged inlet. Sharp edged inlets minimize turbulence just behind the inlet lip; however, this type of inlet is susceptible to flow angle. Hence, a blunt edge is used to decrease this angular sensitivity at the cost of greatly increased turbulent eddies formed downstream of the inlet. Previous inlet designs, like NCAR's community aerosol inlet (CAI) use a shroud in front of a sharp edged inlet to reduce angular sensitivity and turbulence. I have always been puzzled as to why the Falcon inlet did not go to that design, one that is considered superior to the blunt-edged, non-shrouded design. But that is a discussion for another time. In the current study, no mention is made of the fact that even though the inlet is mounted at an angle that

puts it parallel to the nominal flow, it will not always be parallel, particular when the aircraft is flying slower and will have a larger attack angle. Some of the changes in sample efficiency that the authors attribute to TAS may also be related to changes in angle and hence should be taken into account in the analysis. Likewise, how is blunt edge turbulence taken into account in the modeling of the inlet and transport efficiencies? When flows decelerate adiabatically, the temperature rises. This likely has no impact on dust particles but any particle with a volatile coating or composition may lose mass when it heats up. Even if the temperature change is small, this aspect needs to be addressed.

**The aerosol inlet is oriented in forward direction and is slightly tilted towards the nose of the aircraft to account for its average angle of attack ensuring that the apparent wind vector direction is approximately parallel with the inlet's orientation to guarantee near-isoaxial sampling (Walser, 2017). We have analyzed the angle of attack of the Falcon aircraft for the 174 investigated A-LIFE flight sequences: the angle-of attack varies on average by 0.7 degrees over the whole operation range of the Falcon between ground and ~12 km altitude.**

**We would like to clarify that the inlet efficiency is not modelled in this study. It results from the experimentally derived sampling efficiency and the theoretically calculated transport efficiency of the tubing system. We are aware that a number of effects in the inlet system can impact the detected particle number concentration. In the manuscript we will address this point and clarify that the derived inlet efficiency represents the entire inlet system (from the tip of the diffusor to the end of the tube of the isokinetic inlet).**

**Turbulence in the inlet is addressed under comment number 4.**

3.) Table S2 in the supplemental material is confusing since in the main text, the authors state that the out-cabin, UHSAS measurement go out to 1 μm but the Table in the supplement says 125-400 nm. In addition, in the end, it appears that the UHSAS measurements are never actually used since only coarse mode sizes are being analyzed. Then I become further confused with discussion of correcting for drying and fitting 3 modes to the distribution. Why? There are a number of contradictory discussions about the size range of importance. If this study is truly only coarse mode, and most of the figures imply that it is, then please remove all the material that has nothing to do with the coarse mode. In addition, the title should be changed to state explicitly that this is a coarse mode only study.

**The full size distribution used in this study was derived for the A-LIFE campaign with a combination of data from different instruments. For these combined number size distributions (NSD), only a selected size range from each instrument was used (see Table S2 in the Supplement). For example, from the UHSAS-A instrument, we used only the measurements between 125 and 400 nm, although the instrument nominally measures between 60 nm and 1 μm.**

**However, we agree with the Reviewer that the derivation of the aerosol number size distributions for A-LIFE shouldn't be explained in detail in this study and can distract from the actual objective of this study. Section 2.3 is now shortened and only includes a basic overview of the NSD that were derived for the A-LIFE campaign and applied in this study to investigate the depletion and enhancement effects on coarse mode aerosol particles caused by the Falcon inlet as a result of non-isokinetic sampling.**

4.) It is my opinion that it is a mistake to ignore air density, i.e. altitude, in the analysis. Did the authors consider using a non-dimensional parameter that incorporates air density? The Reynolds number would be an excellent option since you are already utilizing other non-dimensional parameters like the Stokes number

**We derived the flow Reynolds number for the first section of the isokinetic inlet (before the flow is split by a flowsplitter, see Figure S01 in the Supplemental Material) with the fitted pressure and temperature values (see Figure S03 and Figure S04 in the Supplement Material) for the true airspeed range used in this study. The calculations were done for the minimum and maximum flow speed inside the isokinetic inlet (indicated with red lines in the following plot). Note, this inlet tube is approximately only 35 cm long before a flowsplitter divides the flow into three sampling lines with laminar flow in each of the sampling lines (see Figure S01 in the Supplement Material).**

**For true airspeeds smaller than approximately 160 m/s, the Reynolds number reveals turbulent flow condition inside the isokinetic inlet (Re > 4000) for the maximum inlet flow of 23.18 m/s. For higher true airspeeds, the Reynolds number becomes smaller indicating transition flow, but never smaller than 2000 (laminar flow). This suggest that at any true airspeed, the flow conditions in the first ~35 cm of the isokinetic inlet tube are never laminar.**

**We will include the plot showing the flow Reynolds number as a function of true airspeed and an explanation in the Supplement Material.**

[Figure]

5.) Line 425 and Fig. 4, This seems an odd choice of markers given that spheres would be more indicative of no dust, since dust is aspherical. But here it is opposite, i.e. triangles are spheres and circles are dust. This bring me to another question about the measurements from the CAS, which am assuming is a CAS-POL. Max, the second author, is an expert with this instrument and yet the polarization capability is not being applied here to separate spherical from aspherical aerosols. This would seem a unique application that only the group at Texas A&M have used. Can the authors explain why they are relying on back trajectories and not direct measurements?

**We agree that the markers used for dust should not be spherical. We will change this in Figure 4.**

**In this study, we based the selection of the 174 investigated measurement sequences on the general A-LIFE aerosol classification scheme which will be described in detail in Weinzierl et al. (2024, in prep.). This classification scheme distinguishes 4 main aerosol types (Saharan**

dust, Arabian dust, mixture with coarse mode, mixture without coarse mode), and further sub-divides each of the main aerosol types into pure, moderately-polluted, and polluted measurement periods. The classification scheme is based on a number of parameters including coarse mode number concentration, black carbon mass concentration, and FLEXPART dispersion simulations. The 174 investigated measurement periods include the aerosol types pure, moderately-polluted, and polluted Saharan and Arabian dust.

Since we wanted to be consistent between different A-LIFE in-situ manuscripts currently in preparation, we decided to base our aerosol classification for all papers on the general aerosol classification scheme and therefore did not use the depolarization signal to distinguish dust from non-dust.

References:

Dollner, M.: Assessment of the global distribution of coarse-mode aerosol and clouds with large-scale in situ aircraft observations, https://doi.org/10.25365/THESIS.72087, PhD thesis, 2022.

Walser, A., Sauer, D., Spanu, A., Gasteiger, J., and Weinzierl, B.: On the parametrization of optical particle counter response including instrument-induced broadening of size spectra and a self-consistent evaluation of calibration measurements, Atmospheric Meas. Tech., 10, 4341–4361, https://doi.org/10.5194/amt-10-4341-2017, 2017.

Walser, A.: On the Saharan Air Layer Aerosol and its Role as a Reservoir of Cloud Condensation Nuclei, PhD thesis, 2017.

Weinzierl B., and Coauthors, in prep.: Investigating mineral dust mixtures in the Eastern Mediterranean: results from the A-LIFE aircraft field experiment, 2024.

**We want to thank Referee #2 for providing valuable comments on our manuscript which helped to improve the manuscript. We will address the comments as described in our following responses in bold.**

This paper analyzes particle size distribution data from aircraft measurements to determine sampling system efficiencies and demonstrates that the combination of both aspiration and transport efficiencies must be considered to accurately determine inlet sampling characteristics. The results presented here are from the A-LIFE field campaign, that had instruments inside and outside aircraft for particle number size distribution and measurements were available over a wide range of altitudes and flight. The paper analyzed data for a forward-facing inlet operated with a narrow band of sampling flowrates that resulted in near isokinetic sampling for a limited range of aircraft speeds (129 to 165 m/s). Downstream of the entrance to the inlet, a small tube was used to sub-sample particles isokinetically from the inlet flow. The authors analyzed the flight data to show that concentrations of coarse particles measured inside the cabin were either depleted or enhanced relative to the ambient at different flight speeds.

This impact of mismatch between sampling and aircraft velocity on the relative concentration of particles in the inlet and the ambient is well known and reasonably well captured by Belyaev and Levin (1970). Efficiencies of transporting particles through sample tubes are moderately well understood using empirical equations though these equations need further vetting with aircraft data such as what is available with the authors here. Considering this background, I'm unsure if this paper presents any new results that add to our knowledgebase on sampling efficiencies. Some of major results, as summarized by the authors in the abstract are: "velocity of the research aircraft has a major impact on the sampling of coarse mode aerosol particles", "larger particles, the enhancement effect at the inlet is still present, but inertial and gravitational particle losses in the transport system get more and more pronounced", and "aerosol particles can either be depleted or enhanced at an aerosol inlet, whereas transport in sampling lines always leads to a loss of particles", and " it is important to consider both, inlet and transport efficiency, when quantifying the sampling efficiency of an aerosol sampling system." All of these findings are well known. I'm not sure what is new in this study.

Considering the lack of any major findings, I recommend rejection of the paper.

**We agree that clarifying the novelty of this study is important. We were not sure to which Belyaev and Levin paper the reviewer is pointing. There is one from 1970 which is only available in Russian language, and another one from 1970 which describes a patent. In Belyaev and Levin (1972, 1974), the authors derived an empirical formula to determine the aspiration efficiency for aerosol inlet systems which was derived for inlet nozzles operated under different ratios between ambient air velocity $U_0$ and air velocity in the inlet tube U (0.17 < $U_0$/U < 5.6). The results of this study can be applied to thin-walled inlet nozzles with external to internal diameters of less than 1.1 (see also Brockmann, 2011).**

**In contrast, our study investigates an aerosol inlet on a fast-flying aircraft considering the entire sampling efficiency of the Falcon aerosol inlet which includes a diffusor, and a 90° bend (see Figure 1 in the manuscript). The sampling efficiency of the aerosol inlet aboard the Falcon research aircraft is characterized for the first time with state-of-the-art in-situ measurements from the A-LIFE field campaign including instruments operated behind the**

Falcon aerosol inlet and mounted at the aircraft wing not affected by the aerosol inlet. So far, only Fiebig (2001) investigated the Falcon aerosol inlet's upper cut-off diameter as a function of altitude. This study was restricted to 6 data points within the planetary boundary layer between 196 m and 2001 m. In contrast, we use a data set of 174 flight sequences to derive a particle size-dependent sampling efficiency as a function of true airspeed (TAS) covering the full TAS range of Falcon operation (and therefore flight altitude between 0 and 12 km).

**In accordance with the Referee #1, we think that this study fits the scope of AMT and that the results are of value for the analysis of past and future studies with the Falcon research aircraft and especially its aerosol sampling system.**

Some minor comments.

Line 143: The reference to isokinetic sampling is a bit confusing. Typically, aircraft isokinetic sampling would be achieved by matching velocities at the tip of the entrance to the diffusor. Here, the sub-sampling tube in the inlet flow is maintained isokinetic. I would recommend that the authors rethink the nomenclature and refer to the isokinetic inlet as isokinetic sub-sampling inlet or tube (or something else that makes clear that the reference to isokinetic sampling is not to the entrance of the inlet that is aspirating ambient flow from outside the aircraft).

**The term "isokinetic inlet" in the context of the Falcon aerosol inlet has an historic background: in Fiebig (2001), the Falcon aerosol inlet was referred to as "isokinetic inlet". Also, in Schneider et al. (2006) it was named "isokinetic probe". However, we want to point out that the inlet flow was never controlled with a bypass which would be necessary to maintain isokinetic sampling conditions for the entire operation range of the Falcon. Isokinetic sampling conditions were met only in a certain range of true airspeed as explained in the manuscript. We will add this clarification to the manuscript.**

Line 159 – It is not clear if the variation in the sample flowrates over the range of 17.87 and 22.83 LPM was intentional or an issue with flow control. Correspondingly it is not clear if the isokinetic inlet velocities of 18 to 23 m/s were controlled to match aircraft speeds over the range of $V\_TAS$ of ~ 129 to 165 m/s or if those values just represented variability in flow control.

**The variation in flowrate between 17.87 and 22.83 lpm results from the Constant Pressure Inlet of the Cloud Condensation Nuclei counter and from the two impactor devices. It is described in more detail in Section 4.2 (Line 743-752), and in the Supplement Material (Section S1, Line 14-29).**

Line 265 – The authors state that they are interested in "understanding the efficiency of the Falcon aerosol inlet for COARSE mode aerosol measurements" and hence the size range of 0.9 to 3 µm is crucial. But this size range is more associated with fine aerosol than coarse.

**There exists a number of definitions for "coarse" particles (e.g. Figure 2 in Adebiyi et al. (2023)). In the revised manuscript, we will add a footnote that coarse mode particles in this study refer to particles > 1 µm in diameter. This definition follows, e.g., Mahowald et al. (2014) and Schumann (2012).**

Line 308: Typo: TSP should be TSI

**Thank you for pointing this out. We will correct this typo in the revised manuscript.**

Line 308: Can the 3760a CPC measure coarse mode particles with high efficiency? What is the upper limit?

**We were in contact with TSI, the manufacturer of the instrument, concerning the data sheet of the TSI 3760a CPC where the upper size limit is given as "> 3 µm". TSI told us that an upper detection limit of > 3 µm means that 3 µm particles are still measured with high efficiency and that the 50% detection efficiency is above 3 µm. In this study, the CPC data were only used to restrict the fits of the size distribution.**

Figure 3: Is it possible to explain the findings at the different sample velocities using the Belyaev and Levin (1970) correlation for anisokinetic sampling. At the lower aircraft velocities (e.g. 112 m/s), the velocity of flow at the inlet entrance is larger than the ambient and hence large particles should be depleted and similarly at the highest aircraft speeds, the particles will be over sampled due to the sub-anisokinetic sampling conditions.

**As mentioned earlier, our study considers a complex sampling system consisting of inlets and sampling lines. In contrast, Belyaev and Levin (1972, 1974) investigated a thin-walled inlet tube.**

As the flow is turned 90 degrees in the sample tube, could inertia deflect particles towards the outside of the bend and hence affect particle concentrations in the central of the flow from where the isokinetic inlet is sub-sampling?

**Yes, this can also affect the particle concentration. We would like to point out that the sampling efficiencies presented in this study were derived by comparing the in-cabin (behind the Falcon inlet) and the out-cabin (not affected by the Falcon inlet) particle number size distribution measured with state-of-the-art in-situ aerosol instrumentation.**

**As mentioned in the manuscript (Line 153-154), our intention was to describe the whole inlet system (from the tip of the entrance of the diffusor to the isokinetic inlet itself which lead to the transport system inside the aircraft) with one transmission efficiency, which contains all of the possible effects that can alter the number concentration of coarse mode aerosol particles. This particle size-dependent transmission efficiency was derived by dividing the experimentally determined sampling efficiency by the theoretically calculated transmission efficiency of the transport system inside the aircraft cabin and is depicted in Figure 8 for multiple true airspeed ranges.**

**References:**

**Adebiyi, A., Kok, J. F., Murray, B. J., Ryder, C. L., Stuut, J.-B. W., Kahn, R. A., Knippertz, P., Formenti, P., Mahowald, N. M., Pérez García-Pando, C., Klose, M., Ansmann, A., Samset, B. H., Ito, A., Balkanski, Y., Di Biagio, C., Romanias, M. N., Huang, Y., and Meng, J.: A review of coarse mineral dust in the Earth system, Aeolian Res., 60, 100849, https://doi.org/10.1016/j.aeolia.2022.100849, 2023.**

Belyaev, S. P. and Levin, L. M.: Investigation of aerosol aspiration by photographing particle tracks under flash illumination, J. Aerosol Sci., 3, 127–140, https://doi.org/10.1016/0021-8502(72)90149-8, 1972.

Belyaev, S. P. and Levin, L. M.: Techniques for collection of representative aerosol samples, J. Aerosol Sci., 5, 325–338, https://doi.org/10.1016/0021-8502(74)90130-X, 1974.

Brockmann, J. E.: Aerosol Transport in Sampling Lines and Inlets, in: Aerosol Measurement, edited by: Kulkarni, P., Baron, P. A., and Willeke, K., John Wiley & Sons, Inc., Hoboken, NJ, USA, 68–105, https://doi.org/10.1002/9781118001684.ch6, 2011.

Fiebig, M.: Das troposphärische Aerosol in mittleren Breiten Mikrophysik, Optik und Klimaantrieb am Beispiel der Feldstudie LACE 98, PhD thesis, Ludwig-Maximilians-Universität, Munich, 2001.

Mahowald, N., Albani, S., Kok, J. F., Engelstaeder, S., Scanza, R., Ward, D. S., and Flanner, M. G.: The size distribution of desert dust aerosols and its impact on the Earth system, Aeolian Res., 15, 53–71, https://doi.org/10.1016/j.aeolia.2013.09.002, 2014.

Schneider, J., Hings, S. S., Nele Hock, B., Weimer, S., Borrmann, S., Fiebig, M., Petzold, A., Busen, R., and Kärcher, B.: Aircraft-based operation of an aerosol mass spectrometer: Measurements of tropospheric aerosol composition, J. Aerosol Sci., 37, 839–857, https://doi.org/10.1016/j.jaerosci.2005.07.002, 2006.

Schumann, U. (Ed.): Atmospheric Physics: Background – Methods – Trends, Springer Berlin Heidelberg, Berlin, Heidelberg, https://doi.org/10.1007/978-3-642-30183-4, 2012.

---

## Referee Report (RR1)

This paper investigates the sampling efficiency of the primary aerosol inlet system used aboard the DLR Falcon aircraft during the A-LIFE field experiment. The authors have combined theoretical calculations with experimental data generated by aerosol particle counters mounted on both the wings of the aircraft and inside its cabin. This research approach is innovative, meaningful and contributes to existing knowledge.

The study puts emphasis on the two well-known constituents of aerosol sampling efficiency; i.e., the aerosol inlet/probe efficiency and the efficiency of the aerosol transport tubing. In addition, it introduces and highlights the importance of using the true air speed of the aircraft in lieu of its flight altitude as a representative figure of merit.

There are inherent uncertainties and limitations, which are correctly identified and adequately discussed by the authors nonetheless. Moreover, although the data analysis is limited to that specific setup and focuses on a relatively narrow range of sampling flow rates and particles of specific composition, the presented study is relevant and of significant interest for the aircraft-based aerosol measurement community. The findings demonstrated may retrospectively be considered and even expanded for the quality control of aerosol measurement data collected by the DLR Falcon and other research aircraft with similar setups.

Overall, the manuscript is very clearly written and demonstrates a credible methodology for the derivation of its findings. The study per se and the quality of presentation are appropriate and well within the scope of AMT. My recommendation is that the manuscript can be published after having been undergone a few minor technical corrections in accordance with the remarks given below.

Please note the following remarks:

Line 154: It would be good to mention the exact angle of the ("slightly tilted") inlet with respect to the fuselage.

Line 329: Explain, if possible, what data the estimated average cabin temperature of 30°C is based upon.

Line 409: Provide a reference or explanation for assuming shape factor $\chi = 1.2$ for mineral dust particles.

Line 751: The syntax of the $V_{TAS}$ condition inside the parentheses is wrong.

Line 796: The term "total volume flow" is used, but the unit in Line 797 is m s$^{-1}$. The same quantity is referred to as "stream velocity" in other parts of the text.

---

## Author Response (AR2)

**We want to thank Referee #3 for the positive feedback on our manuscript and for providing valuable technical corrections. Our replies are given in bold.**

This paper investigates the sampling efficiency of the primary aerosol inlet system used aboard the DLR Falcon aircraft during the A-LIFE field experiment. The authors have combined theoretical calculations with experimental data generated by aerosol particle counters mounted on both the wings of the aircraft and inside its cabin. This research approach is innovative, meaningful and contributes to existing knowledge.

The study puts emphasis on the two well-known constituents of aerosol sampling efficiency; i.e., the aerosol inlet/probe efficiency and the efficiency of the aerosol transport tubing. In addition, it introduces and highlights the importance of using the true air speed of the aircraft in lieu of its flight altitude as a representative figure of merit.

There are inherent uncertainties and limitations, which are correctly identified and adequately discussed by the authors nonetheless. Moreover, although the data analysis is limited to that specific setup and focuses on a relatively narrow range of sampling flow rates and particles of specific composition, the presented study is relevant and of significant interest for the aircraft-based aerosol measurement community. The findings demonstrated may retrospectively be considered and even expanded for the quality control of aerosol measurement data collected by the DLR Falcon and other research aircraft with similar setups.

Overall, the manuscript is very clearly written and demonstrates a credible methodology for the derivation of its findings. The study per se and the quality of presentation are appropriate and well within the scope of AMT. My recommendation is that the manuscript can be published after having been undergone a few minor technical corrections in accordance with the remarks given below.

Please note the following remarks:

Line 154: It would be good to mention the exact angle of the ("slightly tilted") inlet with respect to the fuselage.

**To determine the inclination of the inlet with respect to the fuselage, we got in contact with the DLR department "Flugzeugexperimente". The documentation does not show an inclination of the inlet towards the aircraft's fuselage. The Falcon always flies "with the nose upwards" (for a distribution of angle-of-attack during the SALTRACE campaign with the Falcon, see Figure A2 in Spanu et al., 2020). The direction of flow into the inlet will vary (slightly) with flight condition. There is no indication that this impacts the measurements.**

Line 329: Explain, if possible, what data the estimated average cabin temperature of 30°C is based upon.

**We analyzed temperature measurements inside the inlet tubing of the DMT Cloud Condensation Nuclei Counter (CCNC) instrument which was deployed on the Falcon during A-LIFE. Throughout the A-LIFE aircraft campaign, the temperature at the CCNC inlet was 32.6 ± 3.7 °C on average. The inlet of the CCNC was mounted at the top of the mounting**

**rack, the SkyOPC was mounted in the middle part of the rack where it was slightly cooler. Therefore, we assumed an average temperature of 30°C for the calculations. For clarification we added a sentence to this paragraph in the revised manuscript.**

Line 409: Provide a reference or explanation for assuming shape factor $\chi = 1.2$ for mineral dust particles.

**References were added in Line 409. Furthermore, in Line 329, where the shape factor of mineral dust particles was first mentioned, an explanation was added.**

Line 751: The syntax of the $V_{TAS}$ condition inside the parentheses is wrong.

**Corrected.**

Line 796: The term "total volume flow" is used, but the unit in Line 797 is m s-1. The same quantity is referred to as "stream velocity" in other parts of the text.

**Thank you for spotting this typo. We left the term "total volume flow" in this sentence, but corrected the values to 17.87. – 22.83 l min$^{-1}$.**

**In addition to the requested changes, we also included an additional table (now Table 2) which summarizes the derived cut-off diameters of the sampling system (i.e. combined effect of inlet and sampling lines) from Figure 6 for different particle densities at different altitudes so that interested readers do not have to extract these values from Figure 6.**

**References**

**Spanu, A., Dollner, M., Gasteiger, J., Bui, T. P., and Weinzierl, B.: Flow-induced errors in airborne in situ measurements of aerosols and clouds, Atmospheric Meas. Tech., 13, 1963–1987, https://doi.org/10.5194/amt-13-1963-2020, 2020.**